# Adherence to physical rehabilitation delivered via tele-rehabilitation for people with multiple sclerosis: a scoping review protocol

Geraldine Goldsmith [ID],[1] Jessica C Bollen [ID],[1] Victoria E Salmon [ID],[1] Jennifer A Freeman [ID],[2] Sarah G Dean [ID] [1]

[1]College of Medicine and Health, University of Exeter, Exeter, UK
[2]Faculty of Health, School of Health Professions, University of Plymouth, Plymouth, UK

**Correspondence to**
Geraldine Goldsmith;
geraldine.goldsmith@nhs.net

## ABSTRACT

**Introduction** Using tele-rehabilitation methods to deliver exercise, physical activity (PA) and behaviour change interventions for people with multiple sclerosis (pwMS) has increased in recent years, especially since the SARS-CoV-2 pandemic. This scoping review aims to provide an overview of the literature regarding adherence to therapeutic exercise and PA delivered via tele-rehabilitation for pwMS.

**Methods and analysis** Frameworks described by Arksey and O'Malley and Levac *et al* underpin the methods. The following databases will be searched from 1998 to the present: Medline (Ovid), Embase (Ovid), CINAHL (EBSCOhost), Health Management Information Consortium Database, ProQuest Dissertations and Theses Global, Pedro, Cochrane Central Register of Controlled Trials, US National Library of Medicine Registry of Clinical Trials, WHO International Clinical Trials Registry Platform portal and The Cochrane Database of Systematic Reviews. To identify papers not included in databases, relevant websites will be searched. Searches are planned for 2023. With the exception of study protocols, papers on any study design will be included. Papers reporting information regarding adherence in the context of prescribed therapeutic exercise and PA delivered via tele-rehabilitation for pwMS will be included. Information relating to adherence may comprise; methods of reporting adherence, adherence levels (eg, exercise diaries, pedometers), investigation of pwMS' and therapists' experiences of adherence or a discussion of adherence. Eligibility criteria and a custom data extraction form will be piloted on a sample of papers. Quality assessment of included studies will use Critical Appraisal Skills Programme checklists. Data analysis will involve categorisation, enabling findings relating to study characteristics and research questions to be presented in narrative and tabular format.

**Ethics and dissemination** Ethical approval was not required for this protocol. Findings will be submitted to a peer-reviewed journal and presented at conferences. Consultation with pwMS and clinicians will help to identify other dissemination methods.

## STRENGTHS AND LIMITATIONS OF THIS STUDY

⇒ The SARS-CoV-2 pandemic has resulted in a rapid increase in research into tele-rehabilitation delivery of exercise and physical activity interventions, with this scoping review methodology well suited to the broad and emerging nature of this evidence.

⇒ Decisions regarding study selection, data extraction and quality appraisal will be undertaken by one reviewer, however, to reduce the risk of bias, a second reviewer will check 20% of these decisions.

⇒ The inclusion of critical appraisal of studies will aid the interpretation of results and help to identify gaps within the research and the quality of the evidence base.

⇒ The search strategy includes peer-reviewed evidence from electronic databases alongside organisation websites and conference proceedings.

⇒ Patient and public involvement and engagement was sought in the development of this scoping review, with a plan for further consultation with stakeholders during the review process.

## INTRODUCTION

Adherence is an important predictor of health outcomes[1] with higher adherence levels associated with improved treatment success across a range of healthcare interventions and patient populations.[2–4] Adherence to treatment regimes has the potential to impact healthcare costs; one striking example is where the NHS could save £500 million annually if medication adherence was improved in five key health conditions.[5] This example illustrates the same impact non-adherence may have on other treatment intervention costs, including those relevant to this review in exercise and physical activity (PA) programmes.

Adherence is often used interchangeably with terms such as compliance, participation and concordance,[6] despite the different meanings of these words.[7] The issue is complicated further within therapeutic exercise prescription where various parameters of adherence, that is, the aspects of adherence

that can be measured, have been identified.[7] These parameters include the frequency of exercise (eg, repetitions or sets), the quality of the exercises performed, but also attendance which is only a proxy marker of actually doing the exercises. These authors suggest that while several parameters may be relevant to therapeutic exercise, there is a lack of consensus regarding the relevance and importance in specific contexts.[7]

For clarity and to encompass the multifaceted nature of the concept and parameters of adherence, within this paper the term adherence will be used, as defined by the WHO; 'the extent to which a person's behaviour; taking medication, following a diet, and/or executing lifestyle changes, corresponds with agreed recommendations from a health care provider'[1] (p3). For clarity within this paper, the WHO's definitions of exercise and PA will be used, with the term physical rehabilitation describing physical exercise and PA programmes prescribed with a therapeutic purpose. PA is defined by WHO as 'any bodily movement produced by skeletal muscles that requires energy expenditure'[8] (pvii), and exercise as a subcategory of PA which is 'planned, structured, repetitive, purposeful in the sense that the improvement or maintenance of one or more components of physical fitness is the objective'[8] (pvi). Throughout this paper, the term tele-rehabilitation will refer to the range of technologies which are being used as the method of communication between the rehabilitation professional and patient,[9] allowing for a double communication loop, whereby the patient's performance is monitored and relayed to the clinician who can then respond with appropriate feedback.[10]

Although the beneficial effects of physical rehabilitation are well documented,[11] adherence to such programmes remains problematic with adherence levels to home-based unsupervised exercise reported to be as low as 30%.[12] As adherence is a key predictor of the effectiveness of exercise programmes,[7] there is a clear need to improve adherence to physical rehabilitation programmes to maximise clinical outcomes. However, without a gold standard way to measure adherence to unsupervised exercise[13] and a lack of validated measures,[14] it can be difficult to interpret study results[4]; a poor outcome could be due to poor adherence or a genuine lack of intervention efficacy.

To improve patient adherence to physical rehabilitation, some studies have integrated behavioural change techniques (BCTs) into their programmes.[15–17] BCTs are the active ingredient, or unit of change, within behavioural interventions used to modify a specified behaviour.[15 18] One typical BCT for physical rehabilitation would be setting an exercise behaviour goal.[17] The integration of BCTs into physical rehabilitation programmes has elicited positive results across clinical populations including people living with multiple sclerosis (pwMS).[15–17 19 20]

Multiple sclerosis (MS) is a progressive condition of the central nervous system, affecting over 130 000 people in the UK.[21] Typically, people experience a myriad of sensory, cognitive and motor impairments with subsequent limitations in function,[22 23] which can impact negatively on engagement in exercise and PA.[24] The benefits of exercise for improving symptoms, functional abilities and quality of life for pwMS are well established[25] and physical rehabilitation programmes are frequently prescribed to pwMS.[21] However, pwMS are less physically active than the general population[25] and often experience difficulties in attending outpatient rehabilitation appointments,[26] limiting their access to physical rehabilitation advice and prescription.

Tele-rehabilitation has been proposed as a potential solution for patients who have difficulty in attending outpatient rehabilitation services,[27] and has been promoted within a wider rehabilitation context during the SARS-CoV-2 pandemic[28] with its use likely to continue in clinical practice in line with the Department of Health's Long Term Plan.[29] Within the context of rehabilitation, the 'double communication loop' is an essential element of the tele-rehabilitation as it allows clinicians to adjust or progress a patient's exercise programme according to their performance.[10] When delivering interventions via tele-rehabilitation, the interaction between clinician and patient may be synchronous (in real-time), asynchronous (not in real-time) or mixed, with the feedback received by the clinician delivered online (within the intervention) or offline (with a delay).[10 30]

Within rehabilitation, technology can be used in various contexts including; to enable the communication between the patient and clinician, for example, video conferencing, or as the rehabilitation intervention itself, for example, using wii fit games to improve upper limb motor function. Distinguishing between these two uses of technology is important when attempting to evaluate adherence to physical rehabilitation programmes, as using technology as the rehabilitation intervention may affect adherence.[13 31] This review is interested in exploring whether adherence levels and the effectiveness of BCTs delivered via tele-rehabilitation, within the context of a physical rehabilitation programme, may differ to those delivered in face-to-face settings.

Studies investigating the use of tele-rehabilitation to deliver exercise programmes and interventions to increase PA for pwMS have sought participants'[32–35] and therapists'[33] views on the programmes, however, the authors are not aware of any reviews that have provided an overview of these data. Difficulties with exercise progression were reported by therapists when their communication with participants was conducted via email and online exercise diaries as they were unable to observe participants perform their exercises.[33] Participants have found the tele-rehabilitation delivery acceptable,[33 35] and that it increased flexibility[32–34] as well as reduced the transport and physical energy costs of attending appointments.[32] Systematic reviews have investigated the use of tele-rehabilitation to deliver physical rehabilitation,[36 37] as well as the use of BCTs across tele-rehabilitation and in-person settings for pwMS.[15 19 20] However, these reviews have either; focused on PA levels as the outcome of

interest,[15 19 20] did not explore adherence to the intervention,[36] or included limited information detailing participants' adherence to the prescribed physical rehabilitation programmes.[37] This scoping review aims to address this gap by specifically focusing on detailing the reported level of adherence to physical rehabilitation programmes, use of BCTs and the experiences of pwMS and therapists in adhering to these tele-rehabilitation delivered programmes.

## METHODS AND ANALYSIS

A scoping review methodology is particularly relevant in emerging fields[38] such as tele-rehabilitation; mapping the extent and nature of research and identifying research gaps.[39] The methods developed for this scoping review are based on the six stage framework described by Arksey and O'Malley[39] and Levac et al.[38] This protocol is reported in line with the Preferred Reporting Items for Systematic Reviews and Meta-Analyses Extension for Scoping Reviews.[40]

### Stage 1: identifying the research question

This scoping review aims to provide an overview of the literature regarding adherence to physical rehabilitation delivered via tele-rehabilitation for pwMS. The specific research questions identified are:

1. What levels of adherence are reported by studies prescribing physical rehabilitation delivered via tele-rehabilitation for pwMS?
2. To what degree are valid and reliable measures of adherence used within studies?
3. Is there evidence of integration of BCTs within the physical rehabilitation programmes prescribed, and how well is this reported?
4. Is there any evidence that integrated BCTs have influenced levels of adherence to the prescribed physical rehabilitation programmes?
5. What are the reported experiences of pwMS and physical rehabilitation prescribers regarding adherence to physical rehabilitation?

### Stage 2: identifying relevant studies

Scoping searches were used to identify free text and controlled subject heading terms for the population, intervention and outcome, and a draft search strategy for Medline (Ovid) was developed (strategy 1). As outcomes may not be reported within the title and abstract or picked up by the controlled vocabulary function,[41] a further search strategy (strategy 2) without terms relating to outcomes (ie, adherence) was devised with assistance from an information specialist who provided advice regarding free text terms, appropriate databases and supplementary searching. The two strategies were then compared for duplicates. All papers identified in search strategy 1, were also identified in search strategy 2, however, search strategy 2 identified further relevant papers that were not identified by the first strategy. A search strategy without

terms relating to outcomes will therefore be used within the scoping review to ensure that relevant papers are not missed. Draft search strategies are presented within online supplemental file 1.

The search strategies will be adapted for the following databases: Embase (Ovid), CINAHL (EBSCOhost), Health Management Information Consortium (HMIC) Database, ProQuest Dissertations and Theses Global (PQDT), Pedro, Cochrane Central Register of Controlled Trials, US National Library of Medicine Registry of Clinical Trials, WHO International Clinical Trials Registry Platform portal. Databases will be searched from 1998 to the present day to reflect the start of the scientific publication of tele-rehabilitation studies.[42]

To identify relevant papers not included in bibliographic databases, the following organisation and associated conference websites will be searched: MS Society, MS Trust, Chartered Society of Physiotherapy, UK Society of Behavioural Medicine, International Society of Behavioural Medicine, Open Grey, National MS Society, Rehabilitation in MS and European Committee for Treatment and Research in MS.

A search log will be completed to record searches including the source and dates covered, platform, date of search and number of records yielded. Following study selection, the reference lists of all included studies will be searched for additional relevant papers.

### Stage 3: study selection

Papers reporting information relating to adults (18+) with MS will be included. If mixed populations are included within a study, the study will be included if separate data are reported for adult participants with MS. Information on physical rehabilitation; physical exercises, or PA prescribed for a therapeutic purpose must be included, whether this relates to individual or group delivery, as part of a multifactorial rehabilitation programme or a single intervention. Papers describing tele-rehabilitation provision of the physical rehabilitation programme meeting Laver et al's[9] and Di Tella et al's[10] descriptions will be included, whether the tele-rehabilitation is used as the sole programme delivery method or in combination with another method such as face-to-face.

The review will include papers reporting information relating to the concept of adherence as defined by the WHO.[1] Information relating to adherence may comprise; a method of reporting adherence, adherence levels (eg, exercise diaries, pedometers, questionnaires, measurement scales), investigation of pwMS' or therapists' experiences of adherence or a discussion of adherence.

Study protocols will be excluded, with papers of all other study designs included. In order to minimise language bias, studies in all languages will be eligible with relevant studies translated to English where possible; any that we are unable to translate will be excluded from the review and reason for exclusion noted as language.

Search results will be downloaded into EndNote and duplicates removed. If there are multiple reports of

the same study, these will be compared to ensure that adequate information is obtained and the study's results are only used once. The study selection process will be piloted independently by two team members on 25 studies to check the interpretation of the eligibility criteria and consistency of use. Following the piloting process, the eligibility criteria may be refined.

One reviewer will screen the titles and abstracts of papers, with 20% checked independently by a second reviewer. Papers appearing to meet the eligibility criteria and those where it is unclear from the title and abstract as to whether the criteria have been met will have their full text screened by one reviewer, with 20% screened by a second reviewer. Discrepancies between the two reviewers at any stage of the screening process will be resolved via consensus and discussion with a third person. These processes meet the requirements set out by Plüddemann et al[43] for the study selection stage of restricted systematic reviews and reflect the method used by other authors.[44]

The number of studies excluded at both stages of screening alongside reasons for exclusion will be recorded to enable the completion of a Preferred Reporting Items for Systematic Reviews and Meta-Analyses flow diagram and narrative summary of the screening decision process.

### Stage 4: charting the data
Data extraction will be piloted on five included studies independently by two reviewers, with processes amended as required. This piloting process will ensure that the instructions are applied consistently and the planned data extraction allows relevant characteristics of each study to be presented in the review alongside a quality assessment and details relating to the research questions and review aim.

Following the methodology in other scoping reviews and protocols,[45 46] and meeting the minimum requirements for restricted systematic reviews as set out by Plüddemann et al,[43] data will be extracted into a custom data extraction form by one reviewer. To reduce data bias error, partial verification (20%) of data extraction will be undertaken by a second reviewer.[43] Disagreements will be resolved by consensus and discussion with a third reviewer. The date of extraction and reviewer undertaking the extraction will be recorded alongside key information regarding the paper including: author; year and country of publication; aim and objectives of paper; participant/population information; methods (including study design, blinding, randomisation, time points of data collection); intervention description (including details of tele-rehabilitation methods; synchronous/asynchronous delivery, online/offline feedback, type of technology, exercise or PA interventions, BCTs, comparators); outcomes and results (including adherence measurement tools, eg, session attendance or exercise diaries, information relating to participants' experiences); key findings and discussion points relating to the research questions. Where relevant information is missing from articles, authors will be contacted for further information where possible.

### Stage 5: collating, summarising and reporting the results
This stage will involve three distinct steps; analysis of the data, reporting of the results and outcome in relation to the research questions; and consideration of the meaning of the findings.[38] The methodological quality of included studies will be assessed independently by one reviewer with 20% of decisions checked by a second reviewer, both using tools from the Critical Appraisal Skills Programme suite.[47] A third reviewer will be available to mediate any disagreements.

#### Analysis of the data
The planned analysis of data will occur in two stages. First, an analysis of the data extracted from all included studies with regard to their characteristics and quality appraisal will be undertaken. Second, the research questions will be used to structure and organise an analysis of study findings, outcomes and interventions. Where appropriate, frequency counts will be used for information including study design, use of adherence measures (eg, exercise diary), reporting of adherence levels and intervention characteristics (eg, type of tele-rehabilitation and BCTs used). Data relating to participant adherence levels will be analysed through frequency counts to identify the number of studies reporting a level of adherence and using each measurement tool. Data relating to participants' and prescribers' experiences of adherence within studies will be analysed through the use of categories and themes.

#### Reporting of the results and outcome in relation to the research questions
The characteristics of all included studies incorporating study design, population, interventions and quality appraisal will be presented in a table and narrative summary. Extracted data and the review's findings will be presented in a narrative summary addressing each of the research questions, with the use of tables and diagrams to aid interpretation where appropriate. Data relating to the level of adherence described by studies (research question 1) will be reported with reference to the measurement tool used (research question 2) and adherence parameter measured to provide greater context and appropriate grouping of data relating to adherence levels.

#### Consideration of the meaning of the findings
Findings will be summarised in the context of whether the review's aims have been met and research questions answered. Discussion of the key findings will be included alongside identification of gaps in current knowledge and corresponding implications for future research. The quality assessment of papers may aid the interpretation of results[48] and identification of gaps within the literature.[38]

### Stage 6: consultation
It is planned that consultation will involve pwMS and therapists working with pwMS in community settings (those working with pwMS in their own homes and outpatient settings). The aims of the planned consultation are

to share and discuss preliminary findings, inform future research and develop dissemination methods. The discussion of preliminary findings provides the opportunity to discuss findings within the context of pwMS and clinicians' experiences of adherence to physical rehabilitation delivered via tele-rehabilitation.

## Patient and public involvement and engagement

Patient and public involvement and engagement (PPIE) was sought in the development of the review's aims and research questions alongside the identification of possible dissemination methods to use following the completion of the review. The PPIE was undertaken with a group of four pwMS who provided written feedback on a lay summary of the scoping review protocol. Videoconferencing was then used with three pwMS from this group to explore further their priorities and experience of the use of tele-rehabilitation. This included discussions regarding their general experiences of tele-rehabilitation use alongside specific questions regarding adherence when therapeutic physical rehabilitation programmes have been delivered via tele-rehabilitation.

The feedback regarding the importance of the proposed area of research was useful in shaping the context of the review and its research questions. The PPIE feedback has been incorporated into the protocol through formulation of a research question relating to the experiences of pwMS and identification of dissemination methods as detailed below.

## Ethics and dissemination

Ethical approval was not required in the development of this protocol as the planned methodology involves the review of publicly available data.

The findings of the review will be submitted to a peer-reviewed journal and for presentation at conferences. Further dissemination to clinicians working with pwMS will be guided by the consultation process. PPIE in the design of this protocol identified discussion with MS charities, for example, MS Trust regarding the use of their social media or website platforms and newsletters to promote key findings as an important potential method of dissemination to pwMS, their families and carers. Further PPIE input through the consultation stage of the review may identify further appropriate dissemination methods for pwMS.

**Acknowledgements** PPIE contributors: Thank you to Lynn Tatnell, Jessica Hole and Melanie Guy and other PPIE contributors for their time and help in reviewing the protocol and their contribution to its development. Thank you to PenARC PPIE team for their help and support in arranging the PPIE contributions to this protocol.

**Contributors** Idea for scoping review (GG) with help from JCB, VES, JAF and SGD in its development. Refinement of aims and objectives by GG, JCB, JAF and SGD. Search strategy written by GG with input from PenARC Information Specialist, JCB, VES, JAF and SGD. Methods drafted by GG with input from JCB, VES, JAF, SGD and PenARC search and review clinic team. All authors read and commented on the final draft.

**Funding** This work was supported by Health Education England and the National Institute of Health Research (NIHR) South West Internship funding award (GG). SGD's time is partially supported by the NIHR Applied Research Collaboration South West

Peninsula. The views expressed in this publication are those of the author(s) and not necessarily those of the NIHR or the Department of Health and Social Care.

**Competing interests** GG received funding through an HEE/NIHR ICA Internship (November 2020–April 2021).

**Patient and public involvement** Patients and/or the public were involved in the design, or conduct, or reporting, or dissemination plans of this research. Refer to the Methods section for further details.

**Patient consent for publication** Not applicable.

**Provenance and peer review** Not commissioned; externally peer reviewed.

**ORCID iDs**
Geraldine Goldsmith http://orcid.org/0000-0002-0470-5644
Jessica C Bollen http://orcid.org/0000-0002-6437-5923
Victoria E Salmon http://orcid.org/0000-0002-1536-4750
Jennifer A Freeman http://orcid.org/0000-0002-4072-9758
Sarah G Dean http://orcid.org/0000-0002-3682-5149

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
