## [Reviewer comments · BMJ Open]

ARTICLE DETAILS

TITLE (PROVISIONAL)	Adherence to physical rehabilitation delivered via tele-rehabilitation for people with Multiple Sclerosis: a scoping review protocol
AUTHORS	Goldsmith, Geraldine; Bollen, Jessica; Salmon, Victoria; Freeman, Jennifer; Dean, Sarah

VERSION 1 – REVIEW

REVIEWER	Gallou-Guyot , Matthieu Limoges University, Laboratoire HAVAE
REVIEW RETURNED	21-May-2022

GENERAL COMMENTS	===== GENERAL COMMENTS ===== The main objective of this study is to present the protocol study for a future scoping review assessing the adherence to therapeutic exercise and physical activity delivered through tele-rehabilitation for people with multiple sclerosis. This is an interesting and relevant topic facing the recent lockdowns, and the methodology of the scoping review follows the latest recommendations. I only have minor comments that won't be difficult for the authors to include in their revised version. ===== SPECIFIC COMMENTS ===== ----- MINOR COMMENTS ----- Why did the authors choose a scoping rather than a systematic review design? Why did the authors choose to realise a 10% check for selection of studies and data extraction instead of a fully duplicate analysis? The reference referring to a sentence should be placed before the period or the coma (ex: l. 30, p. 4). Despite the presentation of the used definition, it is not clear if the definition you used of "adherence" (i.e., correspondence with agreed recommendations from a health care provider) includes quantitative (i.g., number of exercise sessions realised, often translated as "participation"; or the number of movements prescribed, often translated as "realisation") but also qualitative (i.g., correctly executed movements, often translated as "compliance") appreciation (l. 40, p. 4). Please precise. The sentence distinguishing telerehabilitation as a communication method and a rehabilitation support a bit confusing to me (l. 43-46, p. 5). Please rephrase.
---

REVIEWER	Di Tella, Sonia
REVIEW RETURNED	22-Aug-2022

GENERAL COMMENTS	This scoping review was conceived to provide an overview of the literature regarding adherence to rehabilitative interventions delivered via tele-rehabilitation for people affected by Multiple Sclerosis. I really appreciated the originality of the topic addressed. However, from a methodological point of view, the work has several critical points. The introduction does not cover the relevant theoretical literature on the topic, and the definition of tele-rehabilitation offered, lacks an up-to-date explanation of the construct that takes into account the double circuit of communication between the patient and the healthcare provider (as Di Tella et al., 2019 DOI: 10.1177/1357633X19850381; Pagliari et al., 2021 DOI: 10.1177/1357633X211054839). According to this literature, the term tele-rehabilitation has been employed “to refer to rehabilitative care beyond the hospital setting in which there was technology that allowed for the double communication loop between the hospital and the patient. [...] the presence of the ‘double loop’ is an essential requirement because of its pivotal role in the planning of individualized patient-centred rehabilitation interventions” (Di Tella et al., 2019). Furthermore, no reference was made to the complexity of the modality in which the telerehabilitation intervention was delivered, i.e. synchronous, asynchronous or mixed, and the specification of the type of monitoring used, online or offline, was also required. Concerning the methods, the choice to entrust screening of all the titles and abstracts of papers by one reviewer and to restrict to 10% of items for evaluation by a second reviewer may have threatened the internal validity of the study. Two reviewers independently should have conducted the whole data extraction and the quality assessment of all selected papers. The too small number of final articles extracted could be affected by this methodological threat and not reflect the actual number of studies in which adherence was reported with self-reported or objective measures (number of sessions completed). Moreover, it is not clear how the relevant literature has been searched. Which were the exact keywords used or which was the role of the information specialist (page 5, line 58)? Overall, this scoping review was not strictly conducted agreeing to the Preferred Reporting Items for Systematic Reviews and Meta-Analyses (PRISMA) guidelines and flow diagram. I recommend authors revise the introduction and methodology and resubmit the article.
---

VERSION 1 – AUTHOR RESPONSE

Reviewer 1 comments:

Comment	Response from Authors
Why did the authors choose a scoping rather than a systematic review design?	The evidence within tele-rehabilitation has expanded quickly over recent years and scoping reviews are the preferred method of review when an emerging area of research needs to be

	comprehensively reviewed. Furthermore, scoping reviews are also suited to identifying knowledge gaps and future research priorities, alongside examining how the research has been conducted (research questions 1 and 2). Systematic reviews are more suited to research questions and aims relating to, for example the effectiveness of an intervention where the body of literature is well established.
Why did the authors choose to realise a 10% check for selection of studies and data extraction instead of a fully duplicate analysis?	In light of the reviewers' concerns regarding the 10% checking of decisions in the selection of studies and data extraction stages, we have increased the percentage of decisions checked by the second reviewer to 20%. This meets the requirements set out by Pludderermann et al (2018). Please see page 7 lines 19-26 and page 7 lines 40-44 for the revisions: One reviewer will screen the titles and abstracts of papers, with 20% checked independently by a second reviewer. Papers appearing to meet the eligibility criteria and those where it is unclear from the title and abstract as to whether the criteria have been met, will have their full text screened by one reviewer, with 20% screened by a second reviewer. Discrepancies between the two reviewers at any stage of the screening process will be resolved via consensus and discussion with a third person. These processes meet the requirements set out by Pluddemann et al [44] for the study selection stage of restricted systematic reviews and reflect the method used by other authors.[45] Following the methodology in other scoping reviews and protocols,[46,47] and meeting the minimum requirements for restricted systematic reviews as set out by Pluddemann et al,[44] data will be extracted into a custom data extraction form by one reviewer. To reduce data bias error, partial verification (20%) of data extraction will be undertaken by a second reviewer.[44]
The reference referring to a sentence should be placed before the period or the coma (ex: l. 30, p. 4).	Thank you for checking the reference convention. We will go through and check that all referencing is in the correct format for this publication
Despite the presentation of the used definition, it is not clear if the definition you used of "adherence" (i.e., correspondence with agreed recommendations from a health care provider) includes quantitative (i.g., number of exercise	Thank you for your comment. In light of this please see the revision below which is now included within the introduction on page 3 lines 30-41:

sessions realised, often translated as "participation"; or the number of movements prescribed, often translated as "realisation") but also qualitative (i.g., correctly executed movements, often translated as "compliance") appreciation (l. 40, p. 4). Please precise.	The issue is complicated further within therapeutic exercise prescription where various parameters of adherence i.e. the aspects of adherence that can be measured, have been identified.[7] These parameters include the frequency of exercise (e.g. repetitions or sets), the quality of the exercises performed, but also attendance which is only a proxy marker of actually doing the exercises. These authors suggest that whilst several parameters may be relevant to therapeutic exercise, there is a lack of consensus regarding the relevance and importance in specific contexts.[7] For clarity and to encompass the multifaceted nature of the concept and parameters of adherence, within this paper the term adherence will be used, as defined by The World Health Organisation (WHO)...
The sentence distinguishing telerehabilitation as a communication method and a rehabilitation support a bit confusing to me (l. 43-46, p. 5). Please rephrase.#	We have rephrased this distinction on page 4 line 46 – page 5 line 4: Within rehabilitation, technology can be utilised in various contexts including; to enable the communication between the patient and clinician e.g. video conferencing, or as the rehabilitation intervention itself e.g. using wii fit games to improve upper limb motor function. Distinguishing between these two uses of technology is important when attempting to evaluate adherence to physical rehabilitation programmes, as using technology as the rehabilitation intervention may affect adherence.[13,32]

Reviewer 2 comments:

Comment	Response from Authors
This scoping review was conceived to provide an overview of the literature regarding adherence to rehabilitative interventions delivered via tele-rehabilitation for people affected by Multiple Sclerosis. I really appreciated the originality of the topic addressed. However, from a methodological	Thank you for taking the time to review our protocol and for your comments.

point of view, the work has several critical points.	
The introduction does not cover the relevant theoretical literature on the topic, and the definition of tele-rehabilitation offered, lacks an up-to-date explanation of the construct that takes into account the double circuit of communication between the patient and the healthcare provider (as Di Tella et al., 2019 DOI: 10.1177/1357633X19850381; Pagliari et al., 2021 DOI: 10.1177/1357633X211054839). According to this literature, the term tele-rehabilitation has been employed "to refer to rehabilitative care beyond the hospital setting in which there was technology that allowed for the double communication loop between the hospital and the patient. [...] the presence of the 'double loop' is an essential requirement because of its pivotal role in the planning of individualized patient-centred rehabilitation interventions" (Di Tella et al., 2019).	Thank you for your comment and links to relevant publications. These have now been cited in the manuscript in relation to the double circuit of communication. In line with your comments we have referred to the double loop of communication on page 4 lines 2-6: Throughout this paper, the term tele-rehabilitation will refer to the range of technologies which are being used as the method of communication between the rehabilitation professional and patient [9], allowing for a double communication loop, whereby the patient's performance is monitored and relayed to the clinician who can then respond with appropriate feedback.[10] In addition, we have made further reference to the double loop of communication on page 4 lines 38-41: Within the context of rehabilitation, the 'double communication loop' is an essential element of the telerehabilitation as it allows clinicians to adjust or progress a patient's exercise programme according to their performance.[10]
Furthermore, no reference was made to the complexity of the modality in which the telerehabilitation intervention was delivered, i.e. synchronous, asynchronous or mixed, and the specification of the type of monitoring used, online or offline, was also required.	We agree that these aspects are important. When we come to undertake the review we will look at these aspects in the resultant included studies. To this end we have added further details relating to the planned data extraction on page 8 lines 3-4 to this protocol manuscript to reflect that these aspects will be examined when the scoping review is undertaken: synchronous/asynchronous delivery, online/offline feedback In addition, reference to the delivery methods of the tele-rehabilitation and type of monitoring has

	been made within the protocol introduction on page 4 lines 41-44: When delivering interventions via tele-rehabilitation, the interaction between clinician and patient may be synchronous (in real time), asynchronous (not in real time) or mixed, with the feedback received by the clinician delivered online (within the intervention) or offline (with a delay).[10, 31]
Concerning the methods, the choice to entrust screening of all the titles and abstracts of papers by one reviewer and to restrict to 10% of items for evaluation by a second reviewer may have threatened the internal validity of the study.	In light of the reviewers' concerns regarding the 10% checking of decisions in the selection of studies, we have increased the percentage of decisions checked by the second reviewer to 20%. This meets the requirements set out by Pluddermann et al (2018) and is in line with methods utilised by other authors. Please see page 7 lines 19-26: One reviewer will screen the titles and abstracts of papers, with 20% checked independently by a second reviewer. Papers appearing to meet the eligibility criteria and those where it is unclear from the title and abstract as to whether the criteria have been met, will have their full text screened by one reviewer, with 20% screened by a second reviewer. Discrepancies between the two reviewers at any stage of the screening process will be resolved via consensus and discussion with a third person. These processes meet the requirements set out by Pluddermann et al [44] for the study selection stage of restricted systematic reviews and reflect the method used by other authors.[45]
Two reviewers independently should have conducted the whole data extraction and the quality assessment of all selected papers. The too small number of final articles extracted could be affected by this methodological threat and not reflect the actual number of studies in which adherence was reported with self-reported or objective measures (number of sessions completed).	As this is a protocol paper there have been no final articles extracted, however we understand reviewer 2's concern. When the review is undertaken, extractors will have a robust document outlining what to include/ exclude and there will be a pilot phase to ensure decisions are congruent between the 2 reviewers, with progression to full screening only when decisions are consistent between reviewers. We have also amended the protocol to increase the percentage of decisions checked by the second reviewer to 20% for both data extraction and quality appraisal, meeting the requirements set out by Pludderman et al (2018). Please see

	page 7 lines 40-44 and page 8 lines 14-15: Following the methodology in other scoping reviews and protocols,[46,47] and meeting the minimum requirements for restricted systematic reviews as set out by Pluddemann et al,[44] data will be extracted into a custom data extraction form by one reviewer. To reduce data bias error, partial verification (20%) of data extraction will be undertaken by a second reviewer.[44] The methodological quality of included studies will be assessed independently by one reviewer with 20% of decisions checked by a second reviewer The JBI guidance for scoping reviews (Peters et al 2020) describes that scoping reviews do not usually include a quality appraisal of included articles, however we have chosen to include this step to aid the interpretation of our results.
Moreover, it is not clear how the relevant literature has been searched. Which were the exact keywords used or which was the role of the information specialist (page 5, line 58)?	Draft search strategies for databases and websites have now been included within the submission. The strategies present the proposed key terms to be included in the search, including free text and controlled subject headings. We have provided clarity regarding the role of the information specialist within the manuscript on page 6 line 12-14: with assistance from an information specialist who provided advice regarding free text terms, appropriate databases and supplementary searching
Overall, this scoping review was not strictly conducted agreeing to the Preferred Reporting Items for Systematic Reviews and Meta-Analyses (PRISMA) guidelines and flow diagram. I recommend authors revise the introduction and methodology and resubmit the article.	Prisma guidelines have been followed as closely as possible, however there is not a specific PRISMA guidance for a scoping review protocol. To this end, the PRISMA extension for scoping reviews has been utilised, however this manuscript is a protocol rather than a completed review, and therefore we have presented as much information as possible towards meeting

	the guidelines including the planned data charting, and collating, summarising and reporting of results. Although we have not been able to “list and define all variables for which data were sought and any assumptions and simplifications made” (Tricco et al 2018), examples of the variables for which data is planned to be sought are given on page 7 final line – page 8 line 7.
--	---

VERSION 2 – REVIEW

REVIEWER	Gallou-Guyot , Matthieu Limoges University, Laboratoire HAVAE
REVIEW RETURNED	01-Feb-2023

GENERAL COMMENTS	GENERAL COMMENTS ===== Authors have considered my comments, and the manuscript appears clearer to me. I only have minor comments that remains and won't be difficult for the authors to include in their revised version. I wish you the best for the conduct of this review. ===== SPECIFIC COMMENTS ===== ----- MINOR COMMENTS ----- The scoping rather than a systematic review design, alongside with the 20% check in duplicate are methodological choices. They are therefore debatable, but not erroneous. I do agree with the reviewer 2: authors should follow the PRISMA guidelines. Conversely, I insist - authors should correct the placement of references all along the manuscript. I don't understand the words used l. 60, p. 6 (“6tilize6ed66”), l. 38, p. 7 (“7tilize7”), l. 32 and 45, p. 8 (“8tilize8ed88n”), and l. 5, 7, 28, 59 p. 9 (“9tilize9”). Please correct.
---

VERSION 2 – AUTHOR RESPONSE

Reviewer 1 comments:

Comment	Response from Authors
Why did the authors choose a scoping rather than a systematic review design?	The evidence within tele-rehabilitation has expanded quickly over recent years and scoping reviews are the preferred method of review when

	an emerging area of research needs to be comprehensively reviewed. Furthermore, scoping reviews are also suited to identifying knowledge gaps and future research priorities, alongside examining how the research has been conducted (research questions 1 and 2). Systematic reviews are more suited to research questions and aims relating to, for example the effectiveness of an intervention where the body of literature is well established.
Why did the authors choose to realise a 10% check for selection of studies and data extraction instead of a fully duplicate analysis?	In light of the reviewers' concerns regarding the 10% checking of decisions in the selection of studies and data extraction stages, we have increased the percentage of decisions checked by the second reviewer to 20%. This meets the requirements set out by Pluddemann et al (2018). Please see page 7 lines 19-26 and page 7 lines 40-44 for the revisions: One reviewer will screen the titles and abstracts of papers, with 20% checked independently by a second reviewer. Papers appearing to meet the eligibility criteria and those where it is unclear from the title and abstract as to whether the criteria have been met, will have their full text screened by one reviewer, with 20% screened by a second reviewer. Discrepancies between the two reviewers at any stage of the screening process will be resolved via consensus and discussion with a third person. These processes meet the requirements set out by Pluddemann et al [44] for the study selection stage of restricted systematic reviews and reflect the method used by other authors.[45] Following the methodology in other scoping reviews and protocols,[46,47] and meeting the minimum requirements for restricted systematic reviews as set out by Pluddemann et al,[44] data will be extracted into a custom data extraction form by one reviewer. To reduce data bias error, partial verification (20%) of data extraction will be undertaken by a second reviewer.[44]
The reference referring to a sentence should be placed before the period or the coma (ex: I. 30, p. 4).	Thank you for checking the reference convention. We will go through and check that all referencing is in the correct format for this publication.
Despite the presentation of the used definition, it is not clear if the definition you used of "adherence" (i.e., correspondence with agreed recommendations from a health care provider)	Thank you for your comment. In light of this please see the revision below which is now included within the introduction on page 3 lines 30-41:

includes quantitative (i.g., number of exercise sessions realised, often translated as "participation"; or the number of movements prescribed, often translated as "realisation") but also qualitative (i.g., correctly executed movements, often translated as "compliance") appreciation (l. 40, p. 4). Please precise.	The issue is complicated further within therapeutic exercise prescription where various parameters of adherence i.e. the aspects of adherence that can be measured, have been identified.[7] These parameters include the frequency of exercise (e.g. repetitions or sets), the quality of the exercises performed, but also attendance which is only a proxy marker of actually doing the exercises. These authors suggest that whilst several parameters may be relevant to therapeutic exercise, there is a lack of consensus regarding the relevance and importance in specific contexts.[7] For clarity and to encompass the multifaceted nature of the concept and parameters of adherence, within this paper the term adherence will be used, as defined by The World Health Organisation (WHO)...
The sentence distinguishing telerehabilitation as a communication method and a rehabilitation support a bit confusing to me (l. 43-46, p. 5). Please rephrase.#	We have rephrased this distinction on page 4 line 46 – page 5 line 4: Within rehabilitation, technology can be utilised in various contexts including; to enable the communication between the patient and clinician e.g. video conferencing, or as the rehabilitation intervention itself e.g. using wii fit games to improve upper limb motor function. Distinguishing between these two uses of technology is important when attempting to evaluate adherence to physical rehabilitation programmes, as using technology as the rehabilitation intervention may affect adherence.[13,32]

Reviewer 2 comments:

Comment	Response from Authors
This scoping review was conceived to provide an overview of the literature regarding adherence to rehabilitative interventions delivered via tele-rehabilitation for people affected by Multiple Sclerosis. I really appreciated the originality of the topic addressed. However, from a methodological	Thank you for taking the time to review our protocol and for your comments.

point of view, the work has several critical points.	
The introduction does not cover the relevant theoretical literature on the topic, and the definition of tele-rehabilitation offered, lacks an up-to-date explanation of the construct that takes into account the double circuit of communication between the patient and the healthcare provider (as Di Tella et al., 2019 DOI: 10.1177/1357633X19850381; Pagliari et al., 2021 DOI: 10.1177/1357633X211054839). According to this literature, the term tele-rehabilitation has been employed "to refer to rehabilitative care beyond the hospital setting in which there was technology that allowed for the double communication loop between the hospital and the patient. [...] the presence of the 'double loop' is an essential requirement because of its pivotal role in the planning of individualized patient-centred rehabilitation interventions" (Di Tella et al., 2019).	Thank you for your comment and links to relevant publications. These have now been cited in the manuscript in relation to the double circuit of communication. In line with your comments we have referred to the double loop of communication on page 4 lines 2-6: Throughout this paper, the term tele-rehabilitation will refer to the range of technologies which are being used as the method of communication between the rehabilitation professional and patient [9], allowing for a double communication loop, whereby the patient's performance is monitored and relayed to the clinician who can then respond with appropriate feedback.[10] In addition, we have made further reference to the double loop of communication on page 4 lines 38-41: Within the context of rehabilitation, the 'double communication loop' is an essential element of the telerehabilitation as it allows clinicians to adjust or progress a patient's exercise programme according to their performance.[10]
Furthermore, no reference was made to the complexity of the modality in which the telerehabilitation intervention was delivered, i.e. synchronous, asynchronous or mixed, and the specification of the type of monitoring used, online or offline, was also required.	We agree that these aspects are important. When we come to undertake the review we will look at these aspects in the resultant included studies. To this end we have added further details relating to the planned data extraction on page 8 lines 3-4 to this protocol manuscript to reflect that these aspects will be examined when the scoping review is undertaken: synchronous/asynchronous delivery, online/offline feedback In addition, reference to the delivery methods of the tele-rehabilitation and type of monitoring has

	been made within the protocol introduction on page 4 lines 41-44: When delivering interventions via tele-rehabilitation, the interaction between clinician and patient may be synchronous (in real time), asynchronous (not in real time) or mixed, with the feedback received by the clinician delivered online (within the intervention) or offline (with a delay).[10, 31]
Concerning the methods, the choice to entrust screening of all the titles and abstracts of papers by one reviewer and to restrict to 10% of items for evaluation by a second reviewer may have threatened the internal validity of the study.	In light of the reviewers' concerns regarding the 10% checking of decisions in the selection of studies, we have increased the percentage of decisions checked by the second reviewer to 20%. This meets the requirements set out by Pluddemann et al (2018) and is in line with methods utilised by other authors. Please see page 7 lines 19-26: One reviewer will screen the titles and abstracts of papers, with 20% checked independently by a second reviewer. Papers appearing to meet the eligibility criteria and those where it is unclear from the title and abstract as to whether the criteria have been met, will have their full text screened by one reviewer, with 20% screened by a second reviewer. Discrepancies between the two reviewers at any stage of the screening process will be resolved via consensus and discussion with a third person. These processes meet the requirements set out by Pluddemann et al [44] for the study selection stage of restricted systematic reviews and reflect the method used by other authors.[45]
Two reviewers independently should have conducted the whole data extraction and the quality assessment of all selected papers. The too small number of final articles extracted could be affected by this methodological threat and not reflect the actual number of studies in which adherence was reported with self-reported or objective measures (number of sessions completed).	As this is a protocol paper there have been no final articles extracted, however we understand reviewer 2's concern. When the review is undertaken, extractors will have a robust document outlining what to include/ exclude and there will be a pilot phase to ensure decisions are congruent between the 2 reviewers, with progression to full screening only when decisions are consistent between reviewers. We have also amended the protocol to increase the percentage of decisions checked by the second reviewer to 20% for both data extraction and quality appraisal, meeting the requirements set out by Pluddeman et al (2018). Please see

	page 7 lines 40-44 and page 8 lines 14-15: Following the methodology in other scoping reviews and protocols,[46,47] and meeting the minimum requirements for restricted systematic reviews as set out by Pluddemann et al,[44] data will be extracted into a custom data extraction form by one reviewer. To reduce data bias error, partial verification (20%) of data extraction will be undertaken by a second reviewer.[44] The methodological quality of included studies will be assessed independently by one reviewer with 20% of decisions checked by a second reviewer The JBI guidance for scoping reviews (Peters et al 2020) describes that scoping reviews do not usually include a quality appraisal of included articles, however we have chosen to include this step to aid the interpretation of our results.
Moreover, it is not clear how the relevant literature has been searched. Which were the exact keywords used or which was the role of the information specialist (page 5, line 58)?	The draft search strategy for Medline has now been included within the submission. The strategy presents the proposed key terms to be included in the search, including free text and controlled subject headings. This strategy will be adapted for other databases and websites, with the draft search strategies for other databases and websites included as a supplementary file for editors. We have provided clarity regarding the role of the information specialist within the manuscript on page 6 line 12-14: with assistance from an information specialist who provided advice regarding free text terms, appropriate databases and supplementary searching
Overall, this scoping review was not strictly conducted agreeing to the Preferred Reporting Items for Systematic Reviews and Meta-Analyses (PRISMA) guidelines and flow diagram. I recommend authors revise the introduction and	Prisma guidelines have been followed as closely as possible, however there is not a specific PRISMA guidance for a scoping review protocol. To this end, the PRISMA extension for scoping reviews has been utilised, however this

methodology and resubmit the article.	manuscript is a protocol rather than a completed review, and therefore we have presented as much information as possible towards meeting the guidelines including the planned data charting, and collating, summarising and reporting of results. Although we have not been able to “list and define all variables for which data were sought and any assumptions and simplifications made” (Tricco et al 2018), examples of the variables for which data is planned to be sought are given on page 7 final line – page 8 line 7.
--

Responses to reviewers’ comments on re-submitted manuscript:

Comment	Responses from authors
Conversely, I insist - authors should correct the placement of references all along the manuscript.	We have amended all references in line with the reviewer’s comments, placing the reference before the period or coma.
I don’t understand the words used l. 60, p. 6 (“6tilize6ed66”), l. 38, p. 7 (“7tilize7”), l. 32 and 45, p. 8 (“8tilize8ed88n”), and l. 5, 7, 28, 59 p. 9 (“9tilize9”). Please correct.	Thank you for highlighting these errors, we have re-checked the manuscript and corrected any errors.